# [2+2] Cyclo-Addition Reactions for Efficient Polymerization on a HOPG Surface at Ambient Conditions

**DOI:** 10.3390/nano12081334

**Published:** 2022-04-13

**Authors:** Lihao Guan, Frank Palmino, Jean-Christophe Lacroix, Frédéric Chérioux, Xiaonan Sun

**Affiliations:** 1Université de Paris, ITODYS, CNRS, 15 rue Jean Antoine de Baïf, 75013 Paris, France; guanlihao666@gmail.com (L.G.); lacroix@univ-paris-diderot.fr (J.-C.L.); 2Université de Franche-Comté, FEMTO-ST, CNRS, UFC, 15B Avenue des Montboucons, 25030 Besancon, France; frank.palmino@univ-fcomte.fr

**Keywords:** scanning tunnelling microscopy, on-surface synthesis, 1D polymers, cyclo-addition

## Abstract

Polymers obtained by on-surface chemistry have emerged as a class of promising materials. Here, we propose a new strategy to obtain self-assembled 1D polymers by using photochemical [2+2] cyclo-addition or by using a mild thermal annealing. All nanostructures are fully characterized by using scanning tunneling microscopy at ambient conditions on a graphite surface. We demonstrated that nature of the stimulus strongly alters the overall quality of the resulting polymers in terms of length and number of defects. This new way is an efficient method to elaborate on-surface self-assembled 1D polymers.

## 1. Introduction

During the past two decades, molecular self-organization has been widely investigated on different kinds of substrates [1,2,3,4,5,6,7,8,9]. More recently, on-surface synthesis was developed as an efficient way to create covalent nanostructures [10,11,12,13,14,15,16,17,18,19,20,21,22,23,24,25,26,27,28,29,30,31]. All these nanostructures have been obtained by bottom-up approaches leading to atomically precise nanostructures, which can be used for great potential applications in molecular electronics, spintronics, energy, catalysis, and other fields. The formation of the surface-confined polymers is mainly governed by the balance between molecule–molecule and molecule–surface interactions, bond strength, reversibility of bond formation, precursor diffusion, and their surface concentration [2,14]. In addition, all on-surface polymerizations require an external stimulus to provoke the initial bond cleavage, which will lead to the polymerization. In the case of on-surface polymerizations, this initiation step is based on thermal annealing, which is the most used stimulus, [10,11,12,13,14,15,16,17,18], on an STM tip-induced reaction [19,20,21,22] or a light illumination [23,24,25,26,27,28]. Among these three approaches, photochemical reactions on surfaces emerge as an elegant solution because they can proceed at room temperatures which can limit the number of defects in synthesized nanostructures, a major weakness of on-surface thermal-induced reactions. Due to their definition, cycloaddition reactions are very promising for on-surface synthesis because there is no side-product, thus avoiding poisoning of the surface [32]. Therefore, on-surface cyclo-addition reactions can be an elegant way to provide extended and undefective covalently bounded nanostructures on unaltered surfaces [24,33,34,35]. However, cyclo-addition reactions follow the Woodward–Hoffman rules, which predict whether a pericyclic process occurs under thermal or photochemical conditions, as well as the stereochemical outcome of the reaction. These rules are based on the conservation of orbital symmetry [36]. Here, we propose to investigate the role of the stimulus to achieve the synthesis of long 1D polymers by activation of C=C bonds on a highly oriented pyrolytic graphite (HOPG) surface at ambient conditions. We propose to answer to several questions: Does the surface impose constraints that favor new mechanism for C=C bond polymerisation? What are the parameters to get the longest 1D-polymers on a surface? In order to fully answer these fundamental questions, we designed specific molecular building blocks, then all adsorbates were fully characterized by scanning tunneling microscopy (STM). We investigated and compared the role of light and thermal annealing as external stimuli to form long 1D-polymers on a HOPG surface. 

## 2. Materials and Methods

### 2.1. Synthesis of 1,4-Bis(4′-vinylphenyl)-2,5-bis(octadecyloxy)benzene (Vinyl-OC18)

All reagents were purchased from Aldrich (Quentin Fallavier, France), except Pd(PPh_3_)_4_, which was purchased from Strem chemical (Bischheim, France), and used as received. The silica gel used for column chromatography was purchased from Merck. The deuterated NMR solvents were purchased from Euriso-top (Saint-Aubin, France). The NMR spectra were recorded using a Bruker AC-300 MHz spectrometer. The procedure is based on two steps [37,38].

2,5-dibromohydroquinone (1 g, 3.8 mmol) and 1-bromo-octadecane (2.45 g, 7.6 mmol) were dissolved in acetone (100 mL). The mixture was heated at reflux for 24 h (Figure 1). The solvent is removed under reduced pressure, then washed three times with 50 mL of water. The resulting yellow powder is recrystallized in 100 mL of ethyl acetate to give the pure 1,4-dibromo-2,5-bis(octadecyloxy)benzene as a white bright solid (Yield: 83%). 

^1^H NMR (400 MHz, Chloroform-d, see Appendix A) δ = 7.09 (s, 2H), 3.95 (t, J = 6.52 Hz, 4H), 1.80 (h, J = 7.92 Hz, 4H), 1.49 (h, J = 7.92 Hz, 4H), 1.25 (m, 56H), 0.87 (t, J = 7.04 Hz, 6H). ^13^C NMR (75 MHz, Chloroform-d) δ = 150.10, 118.50, 111.16, 77.22, 70.34, 31.94, 29.71, 29.67, 29.59, 29.55, 29.37, 29.30, 29.12, 25.94, 22.70, 14.13.

1,4-dibromo-2,5-bis(octadecyloxy)benzene (1.0 g, 1.3 mmol) and 4-vinylphenylboronic acid (0.6 g, 3 mmol) were dissolved in 60 mL of THF (Figure 2). Then, an aqueous solution of K_2_CO_3_ (0.5 M, 6 mL) and Pd(PPh_3_)_4_ catalyst (42 mg, 0.03 mmol) was added. The resulting mixture was heated at reflux for 72 h. Then, the solvent was removed under reduced pressure. The crude solid was dissolved in 100 mL of Chloroform. The organic solution was washed three times with 100 mL of water. The solvent was removed under reduced pressure. The resulting white solid is purified by column chromatography (silica gel, acetone/petroleum ether 1:2). The solid was recrystallized in ethyl acetate to give 1,4-bis(4′-vinylphenyl)-2,5-bis(octadecyloxy)benzene as a pure white solid (Yield: 55%).

^1^H NMR (400 MHz, Chloroform-d, see Appendix A) δ = 7.58 (d, J = 8.24 Hz, 4H), 7.45 (d, J = 8.24 Hz, 4H), 6.98 (s, 2H), 6.79 (dd, J = 10.90 Hz, J = 17.40 Hz, 2H), 5.79 (d, J = 17.40 Hz, 2H), 5.26 (d, J = 10.40 Hz, 2H), 3.90 (t, J = 6.46 Hz, 4H), 1.58 (h, J = 7.92 Hz, 4H), 1.33 (h, J = 7.92 Hz, 4H), 1.25 (m, 56H), 0.87 (t, J = 7.04 Hz, 6H). ^13^C NMR (75 MHz, Chloroform-d) δ = 150.35, 137.93, 136.67, 136.20, 130.45, 129.65, 116.16, 113.63, 77.21, 69.66, 31.94, 29.71, 29.67, 29.59, 29.55, 29.37, 29.30, 29.12, 25.94, 22.70, 14.13.

### 2.2. STM Experiments

Freshly cleaved highly oriented pyrolytic graphite (HOPG from Goodfellow) is used as substrate. Solutions of vinyl-OC18 (Figure 1) molecules in 1-phenyloctane (98%, Sigma Aldrich, Quentin Fallavier, France, concentration of 10^−4^ mol/L) are used for drop-cast deposition. Solutions in DMF (98%, SigmaAldrich, concentration of 10^−4^ mol/L) were prepared for spin coating deposition. The self-assembled networks are obtained at the solid–liquid interface or by drop casting. STM imaging of the samples was performed at the liquid–solid interface or ambient conditions using a SPM Nanoscope-V(Veeco, Bruker, CA, USA) scanning tunneling microscope. Cut Pt/Ir (Goodfellow) tips were used to obtain constant current images at room temperature with a bias voltage applied to the sample. STM images were processed and analyzed using the application FabViewer [39].

## 3. Results

We synthesized 1,4-bis(4′-vinylphenyl)-2,5-bis(octadecyloxy)benzene (vinyl-OC18) molecules, which contain a triphenyl core ended by two vinyl groups and surrounded by two lateral octadecyl chains. The lateral octadecyloxy chains were chosen to promote supramolecular self-assembly by van der Waals interactions between parallel aligned adjacent rows of molecules (interdigitation) and also specific interaction with HOPG surface [3,4]. A triphenyl core contributes to molecule–molecule interactions through π-stacking [5]. The distance between the extremities of vinyl substituents of Vinyl-OC18 molecules is 1.80 nm, while the distance between the two extremities of surrounding lateral octadecyl chains is 5.19 nm (Figure 1).

### 3.1. Supramolecular Self-Assembly on HOPG Surface

Vinyl-OC18 molecules were deposited by drop-casting (20 μL; 10^−4^ mol/L) on HOPG substrate at room temperature or can be generated using the spin coating method. As revealed in STM images, no isolated molecule was observed, but only large extended 2D domains (Figure 2). These 2D domains consist of fused periodic domains, bordered by grain boundaries (Figure 2a).

The repetitive unit of 2D islands consists of parallel pairs of disjoined bright rods (Figure 2b). These repetitive units form a compact periodic network constituted by bright lines, which are separated by darker stripes. The periodicity between the bright lines is 2.9 ± 0.2 nm. In each darker strip, some features (marked by a white dashed ellipse in Figure 2b) are visible and they are aligned along the HOPG (100) orientation at both sides of the bright rods.

Based on our STM observations and measurements, we propose the model of vinyl-OC18/HOPG. Consistent with the features of vinyl-OC18 molecules (Figure 1), each bright rod is attributed to a single aromatic core of a vinyl-OC18 molecule, while the features observed in dark stripes are attributed to lateral C18 alkyl chains of vinyl-OC18 molecules (Figure 2c,d). The unit cell of the supramolecular network vinyl-OC18/HOPG interface is constituted by two elongated-bright protrusions included in a rhomb, as shown by the two vectors U_supra_ and V_supra_ (Figure 2c). The length of these two vectors is U_supra_ = 3.1 ± 0.3 nm and V_supra_ = 1.3 ± 0.1 nm. The angle between these vectors is Θ_supra_ = 120°. The unit cell covering 3.77 nm^2^ for one vinyl-OC18 molecules, the molecular density of the vinyl-OC18/HOPG network is 0.26 molecules per nm^2^.

All the distances measured on STM images are in accordance with the interdigitation of lateral octadecyloxy chains between adjacent molecules. This interdigitation, leading to stabilizing van der Waals interaction between molecules, is widely known to promote the formation of supramolecular self-assemblies onto graphite surfaces [3,4,40]. 

### 3.2. Illumination of Vinyl-OC18 Supramolecular Network on HOPG Surface by UV-Light

We firstly investigated the chemical transformation of vinyl-OC18 supramolecular self-assembly induced by UV irradiation (365 nm) at the solid–liquid interface. The duration of light illumination varied from 20 minutes to 90 min, but no effect of duration was observed. Large scale STM image (Figure 3a) indicates that only a single domain of bright wires is visible. Some rare defects are also visible (surrounded by a plain white circle in Figure 3a). The bright wires are not perfectly linear (as highlighted by the dashed white ellipse in Figure 3a). These bright nanowires are separated by dark stripes (width: 2.9 ± 0.2 nm), including faintly bright features (Figure 3b). 

A high resolution zoomed in STM image on the linear parts of the wires is shown in Figure 3b. We observe clearly that no more bright rods, previously observed in Figure 2b and attributed to single vinyl-OC18 molecules, are visible yet after UV-light exposition. On the contrary, only wires with a continuous brightness over their entire length are observed. The faintly bright features in the dark stripe observed before UV-light illumination (surrounded by a white dashed ellipse in Figure 2b) are still visible after light illumination (Figure 3b). These faintly bright features can be attributed again to lateral octadecyl chains of vinyl-OC18 molecules.

### 3.3. Annealing of Vinyl-OC18 Supramolecular Network on HOPG Surface

We investigated the chemical transformation of vinyl-OC18 supramolecular self-assembly induced by thermal annealing. Hence, a set of experiments consisting of subsequent annealing of the supramolecular network, obtained after drop casting of vinyl-OC18 onto HOPG surface at room temperature, were performed. STM measurement is performed once the DMF solvent is evaporated from thermal treatment. When heated up around 343–363 K, a critical transformation was observed. 

Long-range ordered wire organizations are observed as a single domain (Figure 4a). The size of this single domain is much larger than that observed when vinyl-OC18 molecules were deposited by drop-casting and the perfectly defect-free patterns are distinguished clearly from the previously observed multi-domain self-assemblies (Figure 2a). As observed in the case of UV-light exposition (Figure 4b), 1D bright nanowires are observed uniformly, without any separated rods as those observed before thermal annealing (Figure 2b and Appendix A). The bright nanowires are distanced of 2.9 ± 0.2 nm with the faintly bright features (surrounded by dashed white ellipse in Figure 4b) as previously observed after light illumination (Figure 3b). Such images clearly indicate that vinyl-OC18 molecules are polymerized under thermal treatment.

## 4. Discussion

The STM images obtained after light-induced or thermal-induced polymerization are quite similar. The bright nanowires are aligned along the HOPG(001) orientation. They are separated by dark stripes and the distance between bright parallel nanowires is 2.9 nm. In terms of difference, we only observe that the thermal-induced nanowires are straighter and longer than those obtained by light illumination. Therefore, the nanowires obtained by light illumination or by thermal annealing are structurally identical. However, what the chemical reaction is which leads to the formation of new bright nanowires remains, at that point, unknown.

The chemical transformation of vinyl-OC18 supramolecular self-assembly induced by thermal annealing or by UV-light illumination leads to the complete transformation of starting bright nanorods constituted by separated vinyl-OC18 molecules (Figure 2) into straight bright nanowires (Figure 3 and Figure 4). These chemical transformations only involve the C=C bonds located at the extremities of the vinyl-OC18 molecules. Therefore, we have carefully analyzed the reactions that can involve the C=C bonds. The first reaction is the well-known polyaddition of C=C leading to linear alkyl chains, like in the case of the formation of polystyrene. However, this reaction is impossible in our case because the C=C bonds of two adjacent rows of vinyl-OC18 molecules are separated by a distance of more than 2 nm due to the steric hindrance imposed by lateral octadecyl chains (Appendix A). Besides, such a reaction should lead to aromatic units pending from a linear unsaturated chain. Such a structure would be observed with separated bright spots, which is definitively not observed in the STM images (Figure 3 and Figure 4). A second possible reaction is based on the dehydrogenative homocoupling of two consecutive C=C bonds [41] (Appendix A). Nonetheless, this reaction is only possible on coinage metal surfaces and under ultra-high vacuum, which rules out this possibility in our experimental procedure. Finally, the last possible chemical reaction is of the type [2+2] cycloaddition reaction between two C=C bonds, which leads to the formation of a cyclobutane as linear connecting nodes between consecutive vinyl-OC18 molecules (Figure 5). 

Concerted cycloadditions are one of the most investigated and used chemical reactions. In the case of concerted [2+2] cycloadditions, reactions are allowed by light with a supra-supra symmetry according to the Woodward–Hoffmann rules [36]. Light-induced [2+2] cycloadditions are widely used as efficient synthetic methods in organic chemistry [42]. Therefore, the formation of 1D polymers can be explained by concerted [2+2] cycloadditions between C=C bonds of two consecutive vinyl-OC18 molecules. In the case of UV-light induced concerted [2+2] cycloaddition, the two involved C=C bonds have to be in a supra-supra configuration. The configuration can be achieved by the sliding of vinyl-OC18 molecules relative to each other. Then, the C=C bonds can be rotated in order to be out-of-the-plane of the HOPG surface in order to be parallel. Finally, the concerted cycloaddition occurs to give a cyclobutane as a connecting node between consecutive aromatic cores (Figure 6b and Appendix A). Within this process, the resulting cyclobutane adopts the “puckered” conformation which is the most stable one due to the reduction of the steric hindrance between the substituents [43].

When thermal annealing was used as stimulus, one could explain the formation of 1D polymers by a thermal concerted [2+2] cycloaddition reaction. According to the rule of Woodward–Hoffmann [36], this thermal [2+2] cyloaddition reaction has to be antarafacial on one component. However, the geometrical and orbital constraints ensure that they are encountered only in special circumstances [44]. For instance, concerted thermal [2+2] cycloadditions are mainly encountered in the dimerization of strained alkenes and in (hetero)cumulenes to provide cyclobutanes [45]. In addition, the presence of the underlying HOPG surface strongly limits the antara-supra [2+2] approach because this approach requires a 3D positioning of the C=C bonds which is quite impossible with the presence of the underlying solid surface. Therefore, the thermal [2+2] cyloaddition reaction seems to be quite impossible in our case. 

In order to propose another mechanism to explain the formation of 1D polymer by thermal annealing, we have carefully checked the literature describing thermal dimerization of alkenes in solution. This reaction is rare, and it is based on a two-steps reaction: (i) the cleavage of C=C bonds and the formation of new single C-C bonds and two C-radicals and (ii) the formation of linear C4 chain due to the possibility of hydrogen abstraction from solvent [46]. In our experimental conditions, this process is also possible. The initial dimerization of two C=C bond of two consecutive vinyl-OC18 molecules leads to the formation of the linear biradical C4 chain (Figure 7). However, two different evolutions of the generated biradical can occur. (i) The biradical species could lead to the formation of a cyclobutane by intramolecular cyclisation (Figure 7 right). This mechanism gives the same 1D-polymer as those obtained by UV-light induced concerted [2+2] cycloaddition reaction. (2) These biradicals can undergo H atom transfer from the solvent leading to a linear saturated C4 chain between two adjacent aromatic units. However, we are not able to distinguish the nature of the connecting nodes (linear versus cyclic) in the obtained STM images. Therefore, both mechanisms can explain the observed polymerization.

Experimentally, the 1D-polymers obtained by thermal annealing are without any defect and longer than those obtained by light-induced reactions. This result can be explained by the combination of two parameters which all promote the thermal reaction:(1)The presence of HOPG surface alters the lifetimes and quantum yields of the photoexcited states of the UV-light cycloaddition reactions [47](2)The thermal energy promotes the sliding of vinyl-OC18 molecules.

Finally, on the basis of the experimental data and the physico-chemical discussion, we can provide a model for resulting nanowires observed after UV-Light illumination or thermal annealing (Figure 8). 

The 1D-polymers are all parallel and they are separated by a distance of 2.9 nm due to the interdigitation of lateral octadecyloxy chains between adjacent molecules (i.e., van der Waals interaction between molecules).

## 5. Conclusions

We have successfully polymerized alkene molecules onto a HOPG surface to provide well-organized 1D-polymers. The on-surface polymerization is generated by thermal heating (no solvent), whereas the light irradiation generates polymerization at the solid–liquid interface. We have demonstrated that the reactivity of C=C bonds can be tuned by the presence of a HOPG surface. Indeed, the C=C bonds could be involved in an UV-light induced concerted [2+2] cycloaddition reaction, with respect to the Woodward–Hoffmann rules. In the case of thermal annealing, the polymerization of C=C bonds could give the same polymers that those obtained by light-induced reaction, but it cannot be explained by a concerted cyclo-addition reaction following Woodward–Hoffmann rules. We propose another mechanism, based on surface-assisted two-steps reaction. This thermal induced reaction is very efficient, and it could be used as a new opportunity to provide multi-functional nanostructures. 

## Data Availability

Upon reasonable request to the corresponding authors.

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
