# Peer review of "[2+2] Cyclo-Addition Reactions for Efficient Polymerization on a HOPG Surface at Ambient Conditions"

_nanomaterials, 2022, doi:10.3390/nano12081334_

Round 1

Reviewer 1 Report

The article is devoted to an interesting problem of obtaining functional polymers by the reaction of [2+2]-cycloadditions. The novelty of the work lies in the fact that the authors propose a new approach for obtaining self-organizing polymers on the graphite surface using the [2+2]-cycloaddition reaction. The use of this approach makes it possible to exclude side reactions inherent in [2+2]-cycloaddition in solution, such as radical polymerization or cross-linking processes. In the paper scientifically significant results have been obtained, which are undoubtedly of practical interest. There are some comments questions to the paper.

  1. The experimental part does not provide elemental analysis data and yields of reaction products in the synthesis of 1,4-dibromo-2,5-bis(octadecyloxy)benzene and 1,4-bis(4'-vinylphenyl)-2,5-bis( octadecyloxy)benzene.
  2. The authors use the data of STM experiments as the main evidence for the occurrence of the polymerization reaction by the [2+2]-cycloaddition mechanism. However, according to the solid phase photochemical reaction criteria (GMJ Schmidt, Pure Appl. Chem., 1971, 27, 647–678), pairs of olefin bonds in ligands undergo a [2+2] photocycloaddition reaction if they are aligned in parallel at a distance of 4.2 Å. In this regard, it would be desirable for the authors to provide additional evidence for the occurrence of the polymerization reaction and the formation of cyclobutane units.
  3. Is there any specific role of the graphite surface in the course of self-organizing polymerization?

Author Response

Dear Miona HU,

Thank you very much for forwarding us the referee’s reports on our manuscript submitted for publication in the Nanomaterials Journal. We have appreciated to read the enthusiastic and constructive comments of the referees about our work. As you have suggested, we have seriously considered the different comments and suggestions of the referees. We have made extensive modifications and improvement in the revised manuscript that address the different points raised by the referees in their reports. In the following, we are making point-by-point answers to the referees’ concerns. All significant changes in the manuscript have been highlighted in yellow in the revised version for your convenience. We are confident that this revised manuscript should meet the questions and comments of the referees to their satisfaction.

Sincerely yours,

Dr. F. Chérioux

Referee 1

The article is devoted to an interesting problem of obtaining functional polymers by the reaction of [2+2]-cycloadditions. The novelty of the work lies in the fact that the authors propose a new approach for obtaining self-organizing polymers on the graphite surface using the [2+2]-cycloaddition reaction. The use of this approach makes it possible to exclude side reactions inherent in [2+2]-cycloaddition in solution, such as radical polymerization or cross-linking processes. In the paper scientifically significant results have been obtained, which are undoubtedly of practical interest. There are some comments questions to the paper.

  1. The experimental part does not provide elemental analysis data and yields of reaction products in the synthesis of 1,4-dibromo-2,5-bis(octadecyloxy)benzene and 1,4-bis(4'-vinylphenyl)-2,5-bis( octadecyloxy)benzene.

Author’s reply: The yield of each reaction has been added in the main manuscript and the NMR spectra (1H and 13C) of each molecule are shown as new Figure S1 and S2 in the supporting information. We do not provide elemental analysis because these reactions are well described in the literature and the STM images of the final compounds are in perfect agreement with the expected molecules.

  1. The authors use the data of STM experiments as the main evidence for the occurrence of the polymerization reaction by the [2+2]-cycloaddition mechanism. However, according to the solid phase photochemical reaction criteria (GMJ Schmidt, Pure Appl. Chem., 1971, 27, 647–678), pairs of olefin bonds in ligands undergo a [2+2] photocycloaddition reaction if they are aligned in parallel at a distance of 4.2 Å. In this regard, it would be desirable for the authors to provide additional evidence for the occurrence of the polymerization reaction and the formation of cyclobutane units.

Author’s reply: It is difficult to compare a photochemical reaction at solid phase with those at a solid-liquid or solid-air interface. Indeed, as we mentioned in the main text (p7, lines 253-254) molecules can diffuse at the interface while it is quite impossible in a solid. Therefore, the involved molecules can diffuse until they are close enough to react.

  1. Is there any specific role of the graphite surface in the course of self-organizing polymerization?

Author’s reply: The graphite surface is not playing a significant role for the self-organizing of the polymers as we have observed domains with different orientations after light irradiation (the figure below). And no effect from the three-fold geometry of the HOPG substrate is observed. It does play a role in the self-organization where the lateral alkyls chains and the underlying surface interact. It is more the thermal heating which not only generates polymerization but also optimizes the polymers organizations on HOPG.

The STM is shown in the pdf file.

Reviewer 2 Report

In this paper STM studies are reported in which a monolayer of vinyl-containing monomers are self-assembled at the interface of HOPG and a liquid, the idea being that the 2D self-assembly preorganizes the vinyl moieties for [2+2] cycloaddition reactions. After formation of the monolayer the authors carry out manipulation (thermal or by irradiation) with the purpose to crosslink the vinyl functionalities and form polymer chains via 'on surface' synthesis. They give arguments that the monomers in the layer have polymerized, and support their argumentation with a proposed reaction mechanism. The paper is clearly written and organized. However, I cannot recommend publication of this work due to the following reasons:

  1. My main, overall comment is that I am not at all convinced by the claim by the authors about the occurrence of extensive polymerization reactions (or chemical transformations in general), neither in case of the annealing, nor in case of the thermal treatment. While the quality of the STM images is sufficient, I do not see the formation of “nanowires” (see also my comment 2 below), but rather a thermally induced reorganization of the monomers into a different molecular packing. It might well be that the 2D structures in Fig. 2 represent kinetic ones, and the structures in Figs. 3 and 4 a thermodynamically equilibrated one. I therefore would like to see argumentation by the authors against the possibility of such thermal reorganization as an explanation of their results.
  2. In my opinion the terms “bright wires”, “nanowires” or “nanolines” as used for the structures in the monolayer after irradiation (page 5) and after annealing (page 6) are inappropriate, since for example the observed brightness of the bright signatures in Figs. 3 and 4 is not different from those in Fig. 2. No continuous “lines” are observed. The bright signatures in the STM images in Figs. 3b and 4b show no sign of a ‘wire’-like structure at all. Instead, simply what I see is that the bright submolecular features of the molecules are different amongst the various images.
  3. It is not clear what the effect of prolonged irradiation (> 1 hour) by UV-light is on the local temperature at the interface. In order words: irradiation may be considered as a sort of thermal annealing as well. Related to this: Is the monolayer structure formed by dropcasting dependent on the concentration? Changes in solute concentration at a solid-liquid interface may lead to the generation of different surface polymorphs. Irradiation over prolonged times may lead to local solvent evaporation and thus an increase in solute concentration.
  4. Another argument against interfacial polymer formation would be the formation of very extensive domains as the one shown in Figure 4a, which in my opinion is simply “too perfect” for assemblies of polymeric chains of molecules. Careful inspection of the molecular models superimposed on the STM image in Fig. 2c indicates that the alkenes are certainly not ideally preorganized for [2+2] cycloadditions (too distant from one another). I can believe that at a solid-liquid interface a certain degree of lateral flexibility arises with a better preorganization of the alkenes as the result, however, I find it unlikely that *all* monomers in a 100 x 100 nm domain would become perfectly preorganized in this way. In this respect, it would be good if the authors in some way would be able to produce layers in which “polymerized” and “unpolymerized” molecules were present in one STM image, and in adjacent domains.
  5. I don’t understand the reasoning of the authors on page 8 involving the prerequisites for the thermally induced concerted cycloaddition, i.e. strained alkenes or (hetero)cumulenes. The vinyl monomers in this study do not meet such prerequisites. In addition, I do not understand in what way the underlying HOPG surface could increase the degree of strain in the alkenes.
  6. The proposed mechanism in Fig. 7 does not convince me. The argument of the authors that a HOPG surface can stabilize the proposed highly reactive biradical species is too generalized (the radicals in ref. [47], to which the authors refer, are of a completely different nature of stability). In the present case I would rather expect the possibly formed radicals to react with the underlying graphite, or with residual solvent or water molecules from the ambient.

Other comments:

  1. It is not clearly mentioned whether self-assembly of vinyl-OC18 at a HOPG/1-phenyloctane interface or under ambient conditions (after dropcasting from DMF solution) leads to identical monolayer structures; some clarification is required here.
  2. In section 3.3 on page 5 it is mentioned that after dropcasting of vinyl-OC18 onto HOPG at room temperature no stable self-assembly is observed. This statement however contradicts the STM studies in section 3.1, where the same dropcasting apparently led to stable assemblies. Clarification is required here.
  3. Several reference are made to figures and videos in the supporting information, which I however could not find or download anywhere.
  4. In the synthesis procedures of the two compounds, yields are missing and should be added. In addition, the characterization of these compounds lacks sufficient evidence for their purity and this must be included, either as C,H-elemental analyses, or as a combination of high resolution mass spectrometry and (possibly as supplementary information) hardcopies/scans of the NMR spectra (1H NMR: plot of −1 to 10 ppm, with peak picking and integration; 13C NMR: plot of −10 to 200 ppm, with peak picking)

Author Response

Dear Miona HU,

Thank you very much for forwarding us the referee’s reports on our manuscript submitted for publication in the Nanomaterials Journal. We have appreciated to read the enthusiastic and constructive comments of the referees about our work. As you have suggested, we have seriously considered the different comments and suggestions of the referees. We have made extensive modifications and improvement in the revised manuscript that address the different points raised by the referees in their reports. In the following, we are making point-by-point answers to the referees’ concerns. All significant changes in the manuscript have been highlighted in yellow in the revised version for your convenience. We are confident that this revised manuscript should meet the questions and comments of the referees to their satisfaction.

Sincerely yours,

Dr. F. Chérioux

Referee 2

In this paper STM studies are reported in which a monolayer of vinyl-containing monomers are self-assembled at the interface of HOPG and a liquid, the idea being that the 2D self-assembly preorganizes the vinyl moieties for [2+2] cycloaddition reactions. After formation of the monolayer the authors carry out manipulation (thermal or by irradiation) with the purpose to crosslink the vinyl functionalities and form polymer chains via 'on surface' synthesis. They give arguments that the monomers in the layer have polymerized, and support their argumentation with a proposed reaction mechanism. The paper is clearly written and organized. However, I cannot recommend publication of this work due to the following reasons:

  1. My main, overall comment is that I am not at all convinced by the claim by the authors about the occurrence of extensive polymerization reactions (or chemical transformations in general), neither in case of the annealing, nor in case of the thermal treatment. While the quality of the STM images is sufficient, I do not see the formation of “nanowires” (see also my comment 2 below), but rather a thermally induced reorganization of the monomers into a different molecular packing. It might well be that the 2D structures in Fig. 2 represent kinetic ones, and the structures in Figs. 3 and 4 a thermodynamically equilibrated one. I therefore would like to see argumentation by the authors against the possibility of such thermal reorganization as an explanation of their results.

Author’s reply: We thank the reviewer for the detailed comments. Three STM images are shown below where Figure(a) and (b) are images (exactly the same size) observed before and after the polymerization. We can see clearly in Figure (a) where single molecules are resolved inside the supramolecular stripes. Whereas in Figure (b), there is no separated features which could be attributed to entire monomers, where we call it a nanowire. High resolution in Figure (b) inset image shows continuous DOS observed inside the wires indicating the polymerization. We give an example in Figure (c) where a very similar light generated polymer wires are published in Nature Chemistry, the DOS of the polymer wires is very similar to what is observed in Figure (b) inset image. There are a lot of other articles from STM evidenced polymerization, these kinds of wires are always considered as evidence of chemical transformations.

On another hand, we do not have a STM image from monomer thermal reorganization, due to the fact that as soon as we heated up the sample (above 50˚C to get rid of the solvent), the polymerization occurs. However, I would argue if we assume Figure (b) is a thermal generated reorganization, with such a high resolution, we should have seen the monomers inside the wires, whereas it is not the case. So, we can conclude the wires could only be from polymers.  

  1. In my opinion the terms “bright wires”, “nanowires” or “nanolines” as used for the structures in the monolayer after irradiation (page 5) and after annealing (page 6) are inappropriate, since for example the observed brightness of the bright signatures in Figs. 3 and 4 is not different from those in Fig. 2. No continuous “lines” are observed. The bright signatures in the STM images in Figs. 3b and 4b show no sign of a ‘wire’-like structure at all. Instead, simply what I see is that the bright submolecular features of the molecules are different amongst the various images.

Author’s reply: As we have shown in the figure above, we do have a continuous DOS inside the wires after polymerization. Clear evidence is the DOS image shown in Figure (b) inset image, we will now add this high-resolution DOS image in the main text in Figure4. 

  1. It is not clear what the effect of prolonged irradiation (> 1 hour) by UV-light is on the local temperature at the interface. In order words: irradiation may be considered as a sort of thermal annealing as well. Related to this: Is the monolayer structure formed by dropcasting dependent on the concentration? Changes in solute concentration at a solid-liquid interface may lead to the generation of different surface polymorphs. Irradiation over prolonged times may lead to local solvent evaporation and thus an increase in solute concentration.

Author’s reply: We have varied the light irradiation time between 3 min up to 1.5h to test the efficiency of polymer formation. After 10s of min, the polymerization was efficiently observed. A new image below shows STM image after 3 min irradiation, where two different structures, namely the monomer self-assembly and the light generated polymerization, are observed simultaneously in one STM image. The polymers wires appear much shorter meaning that an elongation of irradiation time is needed to generate more efficient polymerization, which was the images shown in the Figure 2 in main text. The image below is now added in supporting information as Figure S3.

A variation in concentration can indeed sometimes (not always) modify the surface or efficiency of polymerization as reported in the literature. This is however not observed in our cases.

See pdf

STM image (Vs = -0.5V, It= pA, 28 nm x 25 nm) of vinyl-OC18 photo-polymerized (upper part) and unpolymerized (lower part) networks.

  1. Another argument against interfacial polymer formation would be the formation of very extensive domains as the one shown in Figure 4a, which in my opinion is simply “too perfect” for assemblies of polymeric chains of molecules. Careful inspection of the molecular models superimposed on the STM image in Fig. 2c indicates that the alkenes are certainly not ideally preorganized for [2+2] cycloadditions (too distant from one another). I can believe that at a solid-liquid interface a certain degree of lateral flexibility arises with a better preorganization of the alkenes as the result, however, I find it unlikely that *all* monomers in a 100 x 100 nm domain would become perfectly preorganized in this way. In this respect, it would be good if the authors in some way would be able to produce layers in which “polymerized” and “unpolymerized” molecules were present in one STM image, and in adjacent domains.

Author’s reply: We have addressed this point in the previous one.

  1. I don’t understand the reasoning of the authors on page 8 involving the prerequisites for the thermally induced concerted cycloaddition, i.e. strained alkenes or (hetero)cumulenes. The vinyl monomers in this study do not meet such prerequisites. In addition, I do not understand in what way the underlying HOPG surface could increase the degree of strain in the alkenes.

Author’s reply: We agree this comment. We have clarified the corresponding text as following:

In addition, the presence of the underlying HOPG surface strongly limits the antara-supra [2+2] approach because this approach requires a 3D positioning of the C=C bonds which quite impossible with the presence of the so close underlying solid surface. Therefore, the thermal [2+2] cyloaddition reaction seems to be quite impossible in our case. 

  1. The proposed mechanism in Fig. 7 does not convince me. The argument of the authors that a HOPG surface can stabilize the proposed highly reactive biradical species is too generalized (the radicals in ref. [47], to which the authors refer, are of a completely different nature of stability). In the present case I would rather expect the possibly formed radicals to react with the underlying graphite, or with residual solvent or water molecules from the ambient.

Author’s reply: We also agree this comment. We have clarified the corresponding text and modified the Figure 7 in order to describe the alternative way suggested by the referee.

Other comments:

  1. It is not clearly mentioned whether self-assembly of vinyl-OC18 at a HOPG/1-phenyloctane interface or under ambient conditions (after dropcasting from DMF solution) leads to identical monolayer structures; some clarification is required here.

Author’s reply: We have clarified the main text as suggested by the referee.The self-assembled networks are obtained at the solid-liquid interface or by drop casting.” This sentence is now added to page 3.

  1. In section 3.3 on page 5 it is mentioned that after dropcasting of vinyl-OC18 onto HOPG at room temperature no stable self-assembly is observed. This statement however contradicts the STM studies in section 3.1, where the same dropcasting apparently led to stable assemblies. Clarification is required here.

Author’s reply: After drop casting of vinyl-OC18 onto HOPG at room temperature, as the used solvent is DMF (conductive solvent not suitable for STM), STM cannot be performed when there is still low coverage of DMF on surface. We had then obliged to heat up the sample to evaporate all solvent before STM measurement.

The statement of no stable self-assembly is observed after drop casting is inappropriate, we now change it into (page 5, section 3.3) : “after drop casting of vinyl-OC18 onto HOPG at room temperature, STM measurement is performed once the DMF solvent is evaporated from thermal treatment”.

  1. Several references are made to figures and videos in the supporting information, which I however could not find or download anywhere.

Author’s reply: We have uploaded two other files during the submission process: A pdf file corresponding to supporting information and a video to highlight the formation of cyclo-adduct.

  1. In the synthesis procedures of the two compounds, yields are missing and should be added. In addition, the characterization of these compounds lacks sufficient evidence for their purity and this must be included, either as C,H-elemental analyses, or as a combination of high resolution mass spectrometry and (possibly as supplementary information) hardcopies/scans of the NMR spectra (1H NMR: plot of −1 to 10 ppm, with peak picking and integration; 13C NMR: plot of −10 to 200 ppm, with peak picking)

Author’s reply: This point has been also raised by referee 1. We have added NMR spectra in SI.

Round 2

Reviewer 2 Report

The authors have extensively addressed my comments and have included new results in the manuscript and SI. In particular the case in which in the same STM image monomeric species and apparently photopolymerized species are present supports to some extent their claim of polymerization. I am furthermore happy to see that the discussion about the mechanism is more balanced now in comparison to the original manuscript.

Because the authors have toned down their originally somewhat strong claims in the original paper, I am willing to advice acceptance of the work. The creation of well-defined polymerized monolayers composed of custom-made monomers represents an important topic in current-day nanoscience and I feel this work will contribute to the ongoing discussion on this subject.